# The Roles of RUNX Proteins in Lymphocyte Function and Anti-Tumor Immunity

**DOI:** 10.3390/cells11193116

**Published:** 2022-10-03

**Authors:** Wooseok Seo, Aneela Nomura, Ichiro Taniuchi

**Affiliations:** 1RIKEN Center for Integrative Medical Sciences, Laboratory for Transcriptional Regulation, Yokohama 230-0045, Japan; 2Department of Immunology, Graduate School of Medicine, Nagoya University, Nagoya 466-8550, Japan

**Keywords:** RUNX, tumor, lymphocyte, immunity

## Abstract

The Runt-related transcription factor (RUNX) family of proteins are crucial for many developmental and immuno-physiological processes. Their importance in cellular and tissue development has been repeatedly demonstrated as they are often found mutated and implicated in tumorigenesis. Most importantly, RUNX have now emerged as critical regulators of lymphocyte function against pathogenic infections and tumorigenic cells, the latter has now revolutionized our current understandings as to how RUNX proteins contribute to control tumor pathogenicity. These multifunctional roles of RUNX in mammalian immune responses and tissue homeostasis have led us to appreciate their value in controlling anti-tumor immune responses. Here, we summarize and discuss the role of RUNX in regulating the development and function of lymphocytes responding to foreign and tumorigenic threats and highlight their key roles in anti-tumor immunity.

## 1. Introduction

The mammalian RUNX family transcription factors consist of the following three members, each having an essential and unique function: RUNX1 (AML1, CBFA2) is fundamental for embryonic hematopoiesis [1]; RUNX2 (AML3, CBFA1) is a crucial regulator of osteoblast and chondrocyte differentiation [2,3]; and RUNX3 (AML2, CBFA3) is required for the post-natal development of the immune and central nervous systems [4,5]. Deficiency in any of the RUNX family member in mice have detrimental effects on viability, and, hence, the RUNX family are indispensable for sustaining mammalian life. In most cases, the RUNX family members exert their function in transcriptional regulation by heterodimerization with their non-DNA binding partner protein, core binding factor beta (CBFβ), which stabilizes RUNX proteins and strengthens their DNA binding to facilitate transcriptional activity [6,7] (Figure 1). It is also noteworthy that each of the *Runx* genes are transcribed from two alternative promoters, proximal (P2) and distal (P1), giving rise to multiple protein isoforms [8,9]. The patterns of the isoform expression are cell type–specific, time-dependent and nonredundant, and, thus, add another layer of complexity to their functions. For instance, the P2-derived *Runx1* is predominantly expressed during the early stages of embryonic hematopoiesis, whereas P1-derived *Runx1* is increased and takes over P2-derived *Runx1* at late stages of embryonic hematopoiesis and in adult mice. In addition, most of the mature cytotoxic lineage lymphocytes dominantly express P1-derived *Runx3* transcripts [10]. With respect to hematopoiesis, the transforming growth factor-β (TGF-β) is required for hematopoietic stem cell (HSC) quiescence and self-renewal [11]. Notably, RUNX are also activated downstream of TGF-β signaling and cooperate with SMAD signal transducers to regulate the expression of their target genes [12], which is an essential requirement for hematopoiesis and for other important biological systems, as we discuss below.

T cells are lymphocytes of the adaptive immune system that respond to control infectious and tumorigenic threats causing harm to the mammalian host. Conventional TCRαβ^+^ CD4^+^ and TCRαβ^+^ CD8^+^ T cells express the T cell receptor (TCR) which recognizes antigens presented by the respective MHC-II or MHC-I expressing antigen presenting cells (APCs) and promote their activation, clonal expansion and effector function. At the periphery, CD4 and CD8 T cells activate and differentiate into various effector T cells subsets, with each subset having a unique function in the control clearance of viral, fungal, or bacterial pathogens. Activation of CD4^+^ naïve T helper cell requires antigen presentation by MHC-II expressing antigen presenting cells (APCs), which allows their differentiation into multiple effector subsets. The differentiation into each T helper subset is mainly controlled by the cytokines present in their microenvironment. These CD4^+^ T helper subsets include the following: Th1 driven by IL-12: Th2 by IL-4; Th9 by IL-4 and TGF-β; Th17 by IL-6 and TGF-β; Th22 by IL-6 and IL-23; and Treg by TGF-β and IL-2 [13,14]. The cytokines that mediate CD4 T helper differentiation activate the downstream signaling pathways and lead to the upregulation of the master transcription factor (Figure 2). Each of these master transcription factors is responsible for defining transcriptional programs for each T helper subset and finetunes the expression of the appropriate cytokines to control the immune response. CD8 T cells respond to antigens presented on MHC class I protein complexes and differentiate into cytotoxic T lymphocytes (CTL), the effector T cells that express cytolytic proteins and directly mediate the clearance of virally infected cells and malignant cells.

An important feature of T cells is their ability to differentiate into memory cells, a subset of antigen-experienced T cells that survives longer, functions to recall antigens and initiates the secondary immune responses caused by that specific pathogenic microorganism or recurrent tumorigenic cells. Generally, T cells express RUNX1 and RUNX3, as each of them distinctly regulates many aspects of their development and effector function. Briefly, RUNX1 appears to developmentally control the transition of DN3 to DN4 thymocytes through β-selection and RUNX3 controls the differentiation of MHC-I selected DP thymocytes into CD8 T cells [15]. In the context of host immune responses to foreign and tumorigenic cells, RUNX1 and RUNX3 further mediate CD4^+^ and CD8^+^ T cell effector differentiation and function. Lastly, RUNX proteins are valuable players in controlling differentiation of memory and effector T cells and, hence, have great importance in controlling T cell-mediated antitumor immune responses.

The functions of RUNX transcription factors are further implicated in tumorigenesis, as *RUNX* genes are often found with mutations in various types of cancers. RUNX proteins indeed play important roles in tumor biology, but they cannot be simply classified as pro- or anti-tumor proteins. From the perspective of tumor cells, *RUNX* have been reported as both oncogenes and tumor-suppressor genes, which depends on the biological context and cancer types. For instance, somatic mutations in *RUNX1* are highly associated with acute myeloid leukemia (AML) progression with the majority having oncogenic functions [16,17,18]. Another example includes the loss of function mutation in *RUNX3* in gastric cancer patients, which was experimentally determined to be unresponsive to TGFβ signaling and causative for the tumor progression. Further studies in murine gastric epithelium revealed that RUNX3 functions as a tumor-suppressor downstream of TGFβ signaling and forms a protein complex with SMADs to induce the expression of the cyclin-dependent kinase inhibitory protein-1 p21^waf1/cip1^ [19]. CIP1*p21* inhibits cell cycle progression in responses to DNA damage, senescence, growth inhibitory signals, and tumor suppression. RUNX functions in tumorigenesis are further complicated when we consider their roles in the mammalian immune system and functions against tumor progression. Although the roles of RUNX proteins have been extensively studied in immune cell development and differentiation, their roles as regulators of immune responses and their implications in allergic or autoimmune diseases are now becoming established and only a few studies have directly revealed their roles in regulating the immune responses to tumorigenesis (Figure 3). In this review, we summarize recent discoveries on RUNX proteins expressed in lymphocytes, the majority of which have been reported to have prominent roles in anti-tumor activities. Additionally, we discuss the roles of RUNX in regulating lymphocyte function under physiological conditions and raise important questions as to how these immune responses could potentially control the host’s anti-tumor immune responses to cancer.

## 2. RUNX in Th1 and Th-1 like Th17 Function

Among these CD4^+^ T helper subsets, Th1 and Th17 have cytotoxic functions and have potential roles in anti-tumor immunity. Functionally, Th1 cells are crucial for mounting immune responses against intracellular pathogens, whereas Th17 cells mount the host defense against extracellular pathogens. The transcription factors T-bet (encoded by *Tbx21* gene) and RORγt (Retinoic acid-related orphan nuclear receptor) are required for Th1 and Th17 differentiation, respectively. T-bet controls Th1 cell differentiation by activating *Ifnγ* expression whilst suppressing Th2-functional genes, such as *Il4* [20]. Notably, naïve CD4^+^ T cells express high levels of *Runx1* and very low levels of *Runx3*. However, during CD4^+^ T cell activation and Th1 differentiation, *Runx3* expression is elevated, while *Runx1* expression declines [21]. This reciprocal expression pattern of RUNX proteins is dependent on T-bet [22], as both RUNX3 and T-bet cooperate to optimally induce *Ifnγ* expression in Th1 cells and control the Th1 cytotoxic functions. Mechanistically, RUNX3 expression in Th1 cells plays an important role in suppressing the *Il4* gene, through binding to the transcriptional silencer [21,22]. *Runx3* deficiency in murine hematopoietic cells has been linked to the development of various autoimmune diseases, such as colitis [23] and airway hypersensitivity [24]. These significant studies illustrate that T-bet and RUNX transcription factors are critical for Type 1 immune responses mediated by Th1 and NK (ILC1s). Of note, Th1 and NK cells exert T-bet-dependent cytotoxicity via distinct mechanisms, stemming from either adaptive (TCR-dependent) or innate (antigen-independent) immune responses, respectively [25]. Importantly, the Th1 cytokines IL-12 and IFNγ are critically important mediating anti-tumor responses. It will be, therefore, of great interest to examine how *RUNX3* can be experimentally utilized to optimize the proinflammatory functions of Th1 and its infiltration into the tumor microenvironment (TME) to suppress tumor growth.

Th17 cells are mainly responsible for the clearance of extracellular pathogens, such as fungi and bacteria, and secrete the proinflammatory cytokines IL-17 and IL-22. The master transcription factor for Th17-lineage development, RORγt, initiates *Il17a* transcription via binding to ROREs (ROR response elements) [26,27]. RUNX1 is required to induce the transactivation of the *Rorc* locus to execute the Th17-lineage developmental program but this is counteracted by T-bet, through interaction with RUNX1 and blockade of RUNX1-mediated transactivation of *Rorc* expression [28]. Th17 cells can become pathogenic, cause chronic inflammation and are directly involved in the development of experimental autoimmune encephalomyelitis (EAE) in mice and various autoimmune diseases, such as multiple sclerosis and psoriasis in humans [29]. Since chronic inflammation can lead to cancer development, this potentially makes Th17 cells an important contributor to carcinogenesis. Additionally, Th17 cells are frequently identified in human tumors [30,31,32]. However, unlike the clear roles of Th1 cells in regulating anti-tumor immunity, the direct roles of Th17 cells in anti-tumor responses remain elusive. One of the potential reasons why it has been difficult to study Th17 function in anti-tumor immune responses is the fact that the identity of Th17 cells is not developmentally fixed. In contrast to Th1 cells, which have clear distinctiveness in their cell identity, Th17 cells can trans-differentiate to alternative Th subsets or exhibit functional features of Th1 cells. Particularly, Th17 cells often differentiate into IL-17^+^IFN-γ^+^ cells (Th1-like Th17 cells) or IL-17^−^IFN-γ^+^ cells (Th1-like exTh17 cells), which promote the development of most autoimmune diseases [33]. Interestingly, Th1-like Th17 cells co-express RUNX1 and RUNX3 and, together, bind to the ROREs to regulate *Il17a* transcription. This is different to conventional Th1 cells, which specifically express RUNX3 to control T-bet-mediated *Ifnγ* expression [34]. RUNX transcription factors also function together with RORγt to regulate the development of Th1-like Th17 cells, thus hybridizing the Th1 and Th17 transcriptomes and mediating Th1-like Th17 development [34]. In the context of tumorigenesis, Th17 was previously thought to provide a chronic inflammatory environment to accommodate tumor growth, but it is not clear whether Th17 cells play a direct role in modulating the TME [35]. However, the recent discoveries of novel Th17 subsets, including Th1-like Th17 cells, would provide the paradigm shift on the roles of Th17 in tumorigenesis.

## 3. RUNX in CD4 and CD8 CTL

During T cell development, RUNX3 is required for specifying the lineage commitment of MHC-I selected DP thymocytes into CD8^+^ T cells. TCR stimulation of CD8^+^ T cells results in the induction of *T-bet* and *Eomes* at early and late phases of differentiation, both of which promote their differentiation into cytotoxic T lymphocytes (CTLs). Mechanistically, and similarly to that of Th1, RUNX3 cooperates with T-bet to regulate the expression of *Ifng* and *Eomes* in CTL [36]. Eomes is considered as the critical transcription factor for both NK and CD8^+^ CTL cytotoxic functions. Interestingly, T cell cytotoxic functions are not only limited to CD8^+^ T cell lineage. A subset of CD4^+^ T cells can possess the ability to secrete cytotoxic effector molecules, such as IFNγ, perforin and Granzyme B [37]. These so-called CD4^+^ CTLs have been extensively studied and reviewed elsewhere [38,39]. CD4^+^ CTLs function by recognizing antigens presented on MHC-II molecules and destroying these MHCII-expressing APCs. In addition, some virally-infected epithelial [40] or tumorigenic cells [38,41] have been shown to aberrantly express MHCII and are targeted by CD4^+^ CTLs. Since these discoveries are mainly derived from in vitro cytotoxicity assays, the relevance of CD4^+^ CTLs targeting non-APCs under physiological settings are currently being debated. Strikingly, CD4^+^ CTLs are a hybrid of both CD4^+^ helper and CD8^+^ cytotoxic T cells, which, unfortunately, impairs our ability to fully identify the transcriptional regulatory pathways that directly control their development. Moreover, analyses of the cytokine profiles of CD4^+^ CTLs have allowed their further categorization into Th0 CTL, Th1 CTL, Th2 CTL and Th17 CTL, which makes their developmental studies rather complicated [42,43]. Although the exact differentiation pathways are still not clearly defined, recent studies have provided new insights into the origins of CD4^+^ CTLs. Some studies have suggested the possibility of CD4^+^ CTL branching out from a pool of CD4^+^ Th1 cells and the importance of T-bet and RUNX3 in controlling CD4^+^ CTL differentiation [44]. The ThPOK-Runx3 axis was shown to further mediate CD4^+^ CTL differentiation by cross-regulating the CD4^+^ CTL transcriptional program [45]. However, additional studies have reported that CD4^+^ CTL has the potential to differentiate from other CD4^+^ helper lineages, Th2, Th17 or Treg. It is, therefore, likely that there are multiple pathways that contribute to the generation of CD4^+^ CTLs.

Remarkably, CD4^+^ CTLs can also be identified among intestinal TCRαβ^+^ intraepithelial lymphocytes (IELs), a subset of non-conventional T cells found in the epithelial layer of mammalian mucosal linings, particularly in the gut. Rather than expressing the conventional CD8αβ heterodimers, CD4^+^ CTLs in IELs express CD8αα homodimers, but the CD8α subunit itself has no functional significance in the induction of intestinal CD4^+^ CTLs in their intraepithelial environment [46]. Various studies have elegantly revealed that the reciprocal expression of transcription factors RUNX3 and ThPOK are important for CD4^+^ CTL mediated cytotoxicity within IELs [47,48,49]. Unlike secondary lymphoid organs and other peripheral tissues, some intestinal CD4^+^ T cells express modest levels of ThPOK and highly express RUNX3, the latter directly promoting the expression of CTL-signature genes in these intestinal CD4^+^ CTLs. In addition, specific environmental signals, such as TGFβ and RA (Retinoic acid), are responsible for suppressing ThPOK and promoting RUNX3 in these intestinal CD4^+^ CTLs. Moreover, the upregulation of *RUNX3* in intestinal CD4^+^ T cells is important in suppressing the Th17 developmental pathway, and, thus, further driving CD4^+^ CTL development. Furthermore, these studies have used murine colitis models to demonstrate the detrimental loss of *RUNX3* in intestinal CD4^+^ T cells in driving their Th17 polarization and promoting chronic inflammation. This has potential implications for tumorigenesis, as chronic inflammation in the gut can facilitate tissue neoplastic changes and ultimately can promote the progression of colon cancer. Hence, RUNX3 has important roles in restraining the pathogenicity of CD4^+^ CTL cells in tissue inflammation and cancer development. Recent studies have intriguingly shown that CD4^+^ CTL can directly kill tumors aberrantly expressing MHCII by recognizing neoantigens, new antigens encoded by mutated genes and expressed on cancer cells [50,51,52,53]. It has also been shown that activation of the CD137 (TNFSFR9) further contributes to the cytotoxic programming of CD4^+^ CTLs in a RUNX3-dependent way, which has therapeutic potential in treating various human malignancies [54]. Collectively, RUNX3 can strongly influence anti-tumor immunity of CD4^+^ CTLs via direct or indirect pathways.

## 4. RUNX in Tissue Resident memory T Cells (T_RM_)

It has recently surfaced that RUNX3 has vital roles in exerting CTL-mediated anti-tumor immune responses. RUNX3 is recognized as the master transcription factor of CD8^+^ T cell differentiation and function [15], but only a handful of studies have revealed its direct role in CTL-mediated anti-tumor immunity. Resolution of CTL immune responses results in the generation of memory T cells, which provide long-term protection against the same antigenic stimulation and come in 3 subsets. CD8^+^ central memory T (T_CM_) cells continuously circulate between secondary lymphoid organs, while CD8^+^ effector memory T (T_EM_) cells patrol peripheral organs through the blood stream. However, a third subset of non-circulating CD8^+^ memory T cells permanently reside at sites of previously encountered cognate antigens and provide the most adequate immune responses to the antigen they specifically recognize [55,56,57]. These tissue specific CD8^+^ memory T cells are also known as resident memory T cells (CD8^+^ T_RM_) and have great clinical importance to tumor biology. In this context, T_RM_ are identified as critical regulators for cancer–immune equilibrium and protect the host against epidermis-derived cancers [58]. One key extracellular protein T_RM_ that cells exclusively express is integrin αE chain (CD103), which allows CD8^+^ T_RM_ interaction with E-cadherin expressed epithelial cells and promotes tissue retention of T_RM_ cells. [59]. Many efforts are being made to define the specific programs governing CD8^+^ T_RM_ effector functions [60,61] and their tissue residency [62,63,64]. Expectedly, CD8^+^ T_RM_ cells have been recognized as an important player of anti-tumor immunity [58,65,66,67,68]. Recent studies have further shown that CD8^+^ T_RM_ cells have distinct functions with unique transcriptomic programs [69,70].

Specific signals are necessary for the generation of T_RM_, including TGF-β, which plays a critical role in driving full CD103^+^ CD8^+^ T_RM_ maturation in the skin and intestine [71,72], and is specifically dependent on RUNX3, but not RUNX1 function [73]. An experimental study in mice demonstrated that RUNX3 was deemed essential for CD8^+^ T_RM_ cell differentiation and tissue homeostasis [74]. Indeed, *Runx3* expression corelates with the intra-tumoral residency of CD8^+^ T_RM_ cells within human tumors, thus, being a strong indicator of favorable outcomes of various cancers [75,76,77]. Another interesting point drawn from these studies was that both circulating and resident CD8^+^ memory T cells express similar levels of RUNX3, but RNAi-based screening assays in a LCMV infection model revealed RUNX3 is only essential for resident CD8^+^ memory T cells, hence, further demonstrating that RUNX3 has a unique role in controlling T_RM_ development [74]. This elegant study further analyzed effects of *Runx3* loss- and gain-of-function mutations on T_RM_ development and showed its importance in CD8^+^ T_RM_ differentiation and maintenance. Furthermore, it was mechanistically revealed that RUNX3 controls the expression of genes responsible for the tissue-residency of CD8^+^ T_RM_ cells as *Runx3* downregulation severely promoted tumor growth [74]. Therefore, RUNX3 is essential for CD8^+^ T_RM_ differentiation and function against tumorigenesis.

How RUNX3 shapes the transcriptomic programs of CD8^+^ T_RM_ residency remains elusive and further detailed analysis of CD8^+^ T_RM_ cells from multiple peripheral tissues are required. A global analysis of RUNX3-controlled genes in CD8^+^ T_RM_ have revealed that most of the RUNX3-targeted genes were enriched with cell adhesion molecules and various transcription factors [78]. RUNX3 is required to promote chromatin accessibility to genes encoding IRF4 and Blimp transcription factors for CD8^+^ memory T cells, so it will be of great interest to compare the epigenetic landscapes between CD8^+^ T_CM_ and CD8^+^ T_RM_ cells under various tumor settings. It would also be necessary to identify the upstream factors that regulate *Runx3* expression specifically in CD8^+^ T_RM_ cells. Interestingly, one study did identify the Nr4a nuclear receptor family member Nr4a1 in suppressing *Runx3* expression and restraining CD8^+^ T cell development [79], but it was later demonstrated that loss of Nr4a function had no significant effects on *Runx3* expression levels in CD8^+^ TILs (Tumor-infiltrating lymphocytes) [75]. Thus, under physiological settings, *Nr4a* transcription factors have a minor function in regulating *Runx3* expression in CD8^+^ memory T cells. Likewise, it was recently demonstrated that CD4^+^ T cells are also capable of differentiating into CD4^+^ T_RM,_ which is partially dependent on RUNX1 but not RUNX3, for their development [73]. An interesting finding from this study was that enforced expression of both RUNX1 and RUNX3 in CD4^+^ T_RM_ could not fully restore TGFβ-driven T_RM_ programming, demonstrating that RUNX-mediated tissue-residency differs greatly between CD4^+^ and CD8^+^ T_RM_ cells [74]. Therefore, it would be important to understand the mechanistic roles of RUNX proteins in regulating the differentiation and functions of CD4^+^ T_RM_ and CD8^+^ T_RM_ cells. Lastly, the presence of other tissue-resident lymphocytes, such as γδT cells, iNKT (invariant natural killer) cells, and MAIT (mucosal-associated invariant T) cells, in controlling tumorigenesis could also have some influence on the TME, but whether these cells require RUNX for their anti-tumor immune responses is currently unknown.

## 5. RUNX in Regulatory T Cell Function

Regulatory T (Treg) cells play critical roles in controlling immune responses to autoantigens, allergens and tumor antigens and function in suppressing the development of inflammatory and autoimmune diseases. The X chromosome–encoded transcription factor FOXP3 is a lineage-specification factor required for both Treg cell differentiation and for their immunosuppressive functions [80]. FOXP3 controls the expression of *Ctla4* (encoding the immunomodulatory receptor CTLA-4) [81], an immune checkpoint receptor which sequesters CD80/CD86 costimulatory molecules expressed on APCs. The crucial cytokines that drive Treg development and function are TGF-β and IL-2, which are both required to sustain *FOXP3* expression. Since Tregs are essential for maintaining self-tolerance and regulating immune homeostasis, they are problematic in anti-tumor immune responses [82,83]. In the context of tumorigenesis, Tregs suppress effector T cell function via the following mechanisms [84]. Firstly, Tregs secrete the inhibitory cytokines TGF-β and IL-10 to suppress effector T cell function. Secondly, Tregs can induce cytolysis of effector T cells through secretion of granzymes. Thirdly, Tregs can also disrupt effector T cell proliferation and metabolism by transferring the potent inhibitory second messenger cyclic AMP (cAMP) into effector T cells [85]. Fourthly, Tregs express the IL-2 receptor subunit IL-2Rα (CD25), to deplete IL-2 from effector T cells and stop their clonal expansion. Lastly, Tregs expresses CTLA-4 to sequester CD80/86 expressing APCS and stop antigen presentation to CD28-expressing effector T cells. One of the major consequences of the Treg function in tumorigenesis is the promotion of T cell exhaustion, a process which causes T cell dysfunction and elevated expression of inhibitory receptors, including programmed cell death protein 1 (PD-1), lymphocyte activation gene 3 protein (LAG-3), T-cell immunoglobulin domain and mucin domain protein 3 (TIM-3) and CTLA-4 [86]. Current immunotherapies for cancer indeed include CTLA-4 inhibitors, which have been shown to be effective against various cancers, including metastatic melanoma [87].

As mentioned above, RUNX1 and RUNX3 can function under the control of TGF-β signaling, so their roles in Treg differentiation and function were examined. In fact, many studies utilized both murine and human experimental models to demonstrate the requirement of both RUNX3 and RUNX1, to a lesser extent, in promoting both *Foxp3* expression and Treg immunosuppressive activities [88,89,90]. These studies have collectively shown the importance of RUNX-CBFβ complexes in inducing FOXP3 transcription and mediating Treg development and function [91,92,93]. Once expressed, FOXP3 interacts with RORγt and inhibits RORγt-mediated Th17 transcriptional programming, thus sealing the fate of FOXP3^+^ T cells in becoming regulatory T cells. Hence, the reciprocal interaction of FOXP3 and RORγt dictates the differentiation of Treg versus Th17 cells [94]. RUNX1 has been shown to further participate in this pathway by interacting with FOXP3 to halt the Th17 developmental program, but the molecular nature for RUNX1 cooperation with FOXP3 remains unclear [92,95]. Furthermore, many studies have shown the requirement for RUNX proteins in mediating the immunosuppressive activity of Tregs during inflammatory responses [90,92]. However, the roles of RUNX proteins in regulating the function of tumor-infiltrating Tregs or in promoting T cell exhaustion are unknown but should merit further investigation.

## 6. RUNX in ILC Function

In contrast to the activation and functions of CD4 and CD8 T cells, which essentially require antigen-presentation by MHC-II/MHC-I expressing antigen presenting cells (APCs), innate lymphocytes are developmentally primed to rapidly respond and secrete the necessary cytokines and/or proinflammatory proteins required specifically for an immune response. Innate lymphocytes include natural killer (NK) and innate lymphoid cells (ILCs). ILCs are tissue-specific lymphocytes, residing in the mucosal surfaces to provide first line defense against pathogens and facilitate immune responses. Functionally, ILCs are similar (but are not identical) to their CD4 T cell counterparts and it has been suggested that ILCs are the evolutionary precursors of T cells [96]. Based on the cytokines they produce, they have been grouped into ILC1, ILC2 and ILC3, which reflect CD4^+^ Th1, Th2 and Th17 cells, respectively. ILCs first arise from CLPs (Common lymphoid progenitors) in the fetal liver and express the inhibitor of DNA binding (Id2) transcription factor to repress B and T cell developmental pathways. Id2 expressing-CLP can give rise to either NKPs (Natural Killer cell progenitors) or ChILPs (common helper ILC precursors). NKPs develop into NK cells which are guided by Eomes and T-bet transcription factors. On the other hand, ChILPs can differentiate further into various ILC subsets that are specifically controlled by the following transcription factors: T-bet for ILC1; Gata-3/RORγt for ILC2; and RORγt for ILC3. NK cells and ILC1s are immediate responders to viral infections and secrete IFNγ and TNFα. However, what makes NK cells functionally distinct from ILC1s is their ability to produce cytolytic proteins (e.g granzymes and perforin) to mediate cell lysis, making NK cells functionally like CTLs. ILC2s provide the first line of defense against helminths and protect the integrity of the epithelium. ILC2s secrete IL-5 and IL-13 under the influence of IL-33 and TSLP (thymic stromal lymphopoietin) signaling, which initiates type-2 immune responses. Lastly, ILC3s function both in clearance of extracellular parasites and in maintenance of intestinal homeostasis. ILC3s are further grouped into more complex subsets depending on their expression of NCRs (natural cytotoxicity receptors); NKp46 in mice and humans or NKp44 in humans.

RUNX3 has been shown to play important roles in all ILC subsets. Curiously, all subsets of ILC subsets constitutively express RUNX3 [97] but differ in their expression of P1- and P2-derived *Runx3*. Specifically, ILC1s and ILC3s preferentially express the P1-derived *Runx3* and ILC2s express the P2-derived *Runx3*. An interesting point to note is that *Runx3* is deemed essential for ILC1 and ILC3 development, but dispensable for ILC2 development. The roles of RUNX3 further extend into ILC1 and ILC3 homeostasis, as *RUNX3* deficiency in ILC1s and ILC3s results in uncontrollable infection with *C. Rodentim* [97]. It is, however, unknown whether RUNX3 functions in ILC1/3-mediated anti-tumor immune responses; however, all the ILC subsets are detectable within TILs albeit with undefined roles in the tumor microenvironment (TME) [98].

RUNX-CBFβ is not directly involved in ILC2 development, but it does have a key homeostatic function in restraining ILC2 activation in steady-state in the lungs and intestines. In steady state, RUNX-CBFβ antagonizes Gata3 function and suppresses *Il-5* expression [99], the cytokine that drives eosinophil-mediated lung inflammation and allergy. [100]. During Th2-mediated immune responses, RUNX-CBFβ functions to prevent ILC2 exhaustion, a process characterized by reduced Th2 cytokine expression and impaired ILC2 function [99]. Hence, RUNX-CBFβ has a vital role in controlling ILC2 mediated Th2 immune responses. Although RUNX-CBFβ-mediated ILC2 immune responses have not been reported to have direct contributions to anti-tumor immunity, we could speculate that RUNX-CBFβ could indirectly control ILC2 function during anti-tumor immune responses. Notably, ILC2s have integral roles in eosinophil recruitment and activation, and numerous studies have highlighted eosinophils as an integral part of anti-tumor immunity of TME [101]. Thus, an active field of research to treat cancer patients is by boosting eosinophils function for immunotherapy [102]. In addition, ILC2s infiltrate into TME and recruit eosinophils. Hence, optimizing ILC2 function could present a good clinical prognosis in cancer, for instance in human melanomas [103]. One notable function of tumor associated eosinophils is their expression of chemokines, such as CXCX9, CXCL10 and CCL5, which further recruit CD8^+^ CTLs to the tumor sites [104]. CCL5 is an important inflammatory chemokine that is induced at the late phase of inflammation, and we believe that it has pro-cancer functions in tumorigenesis [105]. Additionally, our recent study showed that RUNX3 functions to suppress *Ccl5* gene expression in CD8^+^ T cells, γδT cells and NK cells [105]. Therefore, we speculate that RUNX3 may also function to control the expression of *Ccl5* in eosinophils during inflammation and in TME. Alternatively, ILC2-derived IL-13 could further influence anti-tumor immunity by allowing the migration of activated dendritic cells to the draining lymph nodes [106], which could further facilitate activation of CD8^+^ CTLs. In summary, it would be worth considering the role of the RUNX3-ILC2-eosinophil axis in modulating anti-tumor immune responses, but this will require an intensive investigation. Nevertheless, RUNX3-mediated ILC2 immune responses could potentially be an exciting new area of cancer research for future treatment.

## 7. RUNX in NK Cell Function

CLP-derived NKPs mature into NK cells, and this process is highly dependent on the transcription factors T-bet and Eomes. NK cell activation is influenced by an array of cytokines, such as IL-15 and IL-12. Notably, RUNX3 plays important roles in NK cell development [107], as well as in NK cell activation [108,109,110]. Functional characteristics shared by activated NK and CD8 T cells are that, during antiviral and antitumor responses, they expand clonally, and express the proinflammatory proteins IFNγ, granzyme B, perforin and chemokines to further exacerbate the clearance of the pathogenic cells. Moreover, some NK and CD8 T cells differentiate into memory lymphocytes [111], the key immune-experienced cells that rapidly respond to secondary immune responses. Thus, NK and CD8 T cells have significant contributions to antitumor immunity [112,113]. During NK cell activation, the signal and activated transducer 4 (STAT4) transcription factor (activated downstream of by IL-12 signaling) induces the expression of both *Runx1* and *Runx3*, which, in turn, contribute to NK cell expansion, survival and memory formation [114]. Interestingly, it was recently shown that *Runx2* also has a significant role in human NK cell differentiation [110], all of which collectively suggest that every RUNX member is important in NK biology. Lastly, our recently published research further enhanced our current understanding of NK cell-mediated anti-tumor immunity. We uncovered a novel anti-tumor role of RUNX3 in suppressing *Ccl5* expression in lung-resident NK cells [105]. Thus, RUNX3 is fundamental for suppressing melanoma metastasis to secondary tissues, but the functional roles of RUNX proteins in tumor-associated NK cells require further investigation. Overall, RUNX-mediated regulation of NK cell differentiation and activity exerts critical influences on anti-tumor immunity.

## 8. Conclusive Remarks

Our discussion on the roles of RUNX transcription factors in regulating the mammalian immune system should convince our readers how valuable RUNX proteins are in controlling anti-tumor responses. Current studies have, overall, identified RUNX3 as the dominant member in controlling NK- and CD8^+^ CTL-mediated antitumor immune responses, but whether RUNX3 is also implicated in controlling the functions of tumor-associated ILCs and myeloid cells is obscure. In addition, the exact roles of RUNX in controlling the immunosuppressive function of tumor-infiltrating regulatory T cells remains unknown. These unresolved questions, thus, burden our view in fully understanding how RUNX regulates tumorigenesis and anti-tumor responses. Our take-home message is that our understanding of the mammalian immune system is constantly evolving. We can no longer apply dogmas or paradigms to understand in vivo phenomena because the limitations of concepts of so-called “master transcriptional regulator” or “signature cytokines” in understanding complicated immunological responses have been shown. For example, we are now confronted by many new and novel subsets of immune cells, such as IL-9-producing CD8^+^ T cells [115] or IL-17-producing γδ T cells [116]. Furthermore, study on immunology has further evolved by incorporating plastic epigenetic changes rather than snapshots of transcriptomics, as well as cellular heterogenicity at single-cell level rather than population-based approaches [117]. This, however, should not underestimate the fact that transcription factors are the crucial taskforces of cellular development and differentiation. Therefore, immune cells must respond to tumor cells by translating signals received from the TME into transcriptional networks. Thus, technical advances in the future will allow us to fully understand how RUNX proteins shape the cytotoxicity of immune cells against tumors.

## Figures and Tables

**Figure 1 cells-11-03116-f001:**
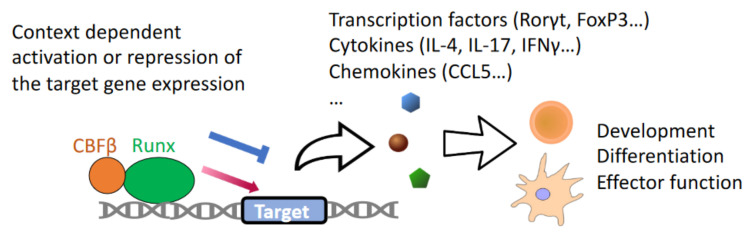
All 3 RUNX proteins (RUNX1, RUNX2 and RUNX3) dimerize with CBFβ to stabilize RUNX and mediate RUNX DNA binding activity. RUNX-CBFβ complexes bind to *cis*-regulatory elements in their target loci and activate/repress the expression of target genes which mainly encode various transcription factors, cytokines and chemokines. The transcriptional regulation of RUNX these target genes collectively function to regulate the development, differentiation and function of immune cells.

**Figure 2 cells-11-03116-f002:**
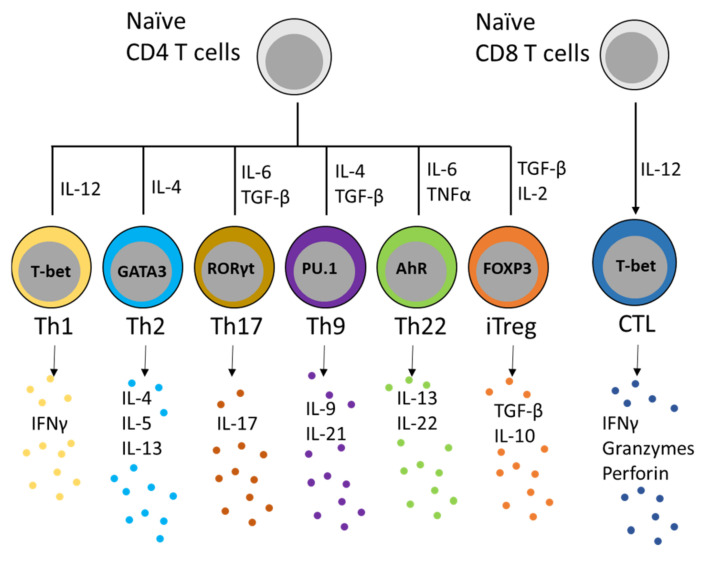
Activation of CD4+ naïve helper T cells by antigen stimulation and cytokine signaling promotes their differentiation into multiple effector subsets, which is developmentally controlled by the master transcription factors as indicated. These CD4+ T helper subsets include: Th1 driven by IL-12 and T-bet; Th2 by IL-4 and GATA3; Th17 by IL-6, TGF-β and RORγt; Th9 by IL-4, TGF-β and PU.1; Th22 by IL-6, TNFα and AhR (Aryl hydrocarbon receptor); and iTreg by TGF-β, IL-2 and FOXP3. For CD8+ T cytotoxic cells, IL-12 and T-bet play an important role in inducing their differentiate into CTL (cytotoxic T lymphocytes). For some cases, it should be noted that under physiological conditions, the cells that produce the cytokines to mediate effector T cells differentiation are still not clearly defined. The master transcription factor(s) for each T cell effector subset defines their transcriptional program and promotes the expression of the signature effector cytokines (as indicated below each T cell subset) to control the host immune responses to pathogenic and malignant cells.

**Figure 3 cells-11-03116-f003:**
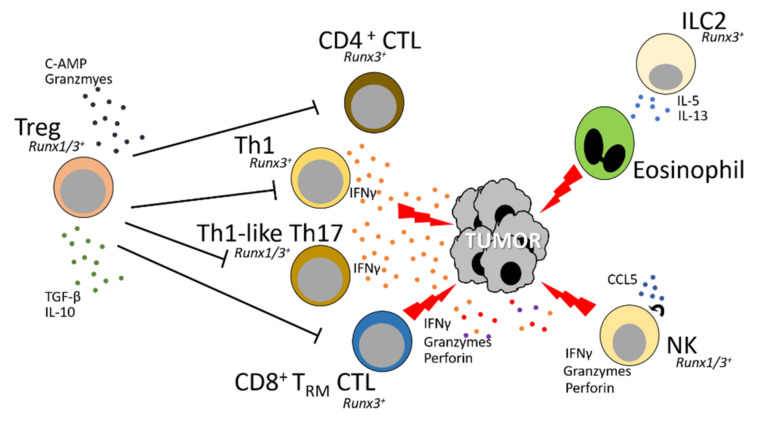
RUNX proteins directly regulate the function of various effector immune cells, including ILC2, NK, Th1, Th1-like Th17, CD4^+^ CTL, CD8^+^ T_RM_ CTL and Tregs, all of which infiltrate into tumors and play important roles in anti-tumor immunity. These tumor-infiltrating lymphocytes (TILs) can have direct effects on the residency and cytolytic functions of CD4 CTL and CD8^+^ T_RM_ CTL cells. Additionally, RUNX proteins can regulate anti-tumor immunity by regulating the expression of chemokines or cytokines in NK and Th1-like Th17 (or Th1) cells and promote their anti-tumor functions. Furthermore, RUNX-mediated ILC2s can function to regulate the recruitment of eosinophils into the tumor microenvironment (TME) and boost the clearance of tumorigenic cells.

## Data Availability

Not applicable.

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
