# Peer review of "The Roles of RUNX Proteins in Lymphocyte Function and Anti-Tumor Immunity"

_cells, 2022, doi:10.3390/cells11193116_

Round 1
Reviewer 1 Report
This review paper comes from the laboratory of Dr. Taniuchi, an expert in transcriptional regulation in immune system. It covers the role of RUNX proteins in various cell types involved in anti-cancer immunity. Overall, the review contains good description of lymphocyte function and anti-tumor immunity, though immunology at times prevails over RUNX function review. Only minor changes are suggested below:
1. “the RUNX family members can only exert their function in transcriptional regulation by heterodimerization with their non-DNA binding partner protein, core binding factor beta.” -Is this really true? The Huang paper only talks about protection from degradation…
2. When talking about the roles of RUNXs in tumorigenesis, it may be a good idea to mention translocations.
3. Page2 “Generally, T cells express RUNX1 and RUNX3, as each of them distinctly regulates every aspect of their development and effector function. “ - saying every aspect is quite ambitious..
4. Page3 Figure 2 has minor issue in the top left corner
5. Page4 “RUNX1 is required to induce the transactivation of Rorgt locus to execute the Th17-lineage developmental program but this is counteracted by T-bet, through inter- action with RUNX1 and blockade of RUNX1-mediated transactivation of Rorgt expression (Lazarevic et al. 2011).” – locus name is Rorc, RORγt – is protein
6. Page4 “Additionally, Th17 cells are frequently identified in human tumors (Zou and Restifo 2010; Vitiello and Miller 2020; Bailey et al. 2014).but, unlike the clear roles of Th1 cells in regulating anti-tumor immunity, the direct roles of Th17 cells in anti-tumor responses remain elusive. “
7. Page4 “One of the potential reasons why it has been difficult to study Th17 function in anti-tumor immune responses is the fact that the individuality of Th17 cells is rather plastic”. – the term “cells individuality” is not clear
8. Page4 “Interestingly, Th1-like Th17 cells co-express RUNX1 and RUNX3 and they both bind to the ROREs in a T-bet-dependent manner to regulate Il17a transcription, which is in converse to that in Th1 cells, which specifically express RUNX3 to control T-bet-mediated Ifnγ expression (Wang et al. 2014) “– the sentence is very bulky and a bit confusing
9. The last paragraph of part2 on the page4 is predominantly about Th17 biology and concludes the role of Th17 subset in tumorogenesis rather than about RUNX role. In the similar manner Page 5 paragraph one mostly discusses CTL cells origin and just a few words about RUNX. Same for Treg and ILC chapters.
10. Page5 “Some studies have specifically pointed out the possibility of CD4+ CTL to branch out from a pool of CD4+ Th1 cells and have shown the importance of T-bet and RUNX3 in controlling CD4+ CTL differentiation (Takeuchi et al. 2016) as well as the involvement of ThPOK-Runx3 axis in further mediating this process (Serroukh et al. 2018)”. – this huge sentence can be split by two.
11. Page5 “Collectively, the RUNX3 transcription factor can strongly influence anti-tumor immunity of CD4+ CTLs via direct or indirect pathways. “
12. Page5 “Resolution of CTL immune responses results in the generation of immunological memories, in the forms of memory T cells, which provide a long-term protection against the same antigenic stimulation and come in 3 subsets. “
13. Page6 “Furthermore, it was mechanistically revealed that RUNX3 controls the expression of genes responsible for the tissue-residency of CD8+ TRM cells as Runx3 downregulation severely promoted tumor growth”. – reference?
14. Page7 RUNX1 has been shown to further participate in this pathway by interacting with FOXP3 to halt the Th17 developmental program, but the molecular mechanism for RUNX1 cooperation with FOXP3 remains unclear (Zhang, Meng, and Strober 2008) - Ono et al 2007 showed by co-immunoprecipitation that RUNX1 physically interacts with FOXP3.
15. The review lacking recent information about RUNX as a prognostic marker in tumor infiltration https://pubmed.ncbi.nlm.nih.gov/34485309/
https://pubmed.ncbi.nlm.nih.gov/36092942/
The following phrases are awkward and need to be re-worded:
Deficiency in any of the RUNX family member have detrimental effects on mortality,
These pro-founding studies illustrate
Author Response
<To the Reviewer 1>
Overall, the review contains good description of lymphocyte function and
anti-tumor immunity, though immunology at times prevails over RUNX function review.
We thank this reviewer for the understanding the scope of our review. We believe that many Runx reviews redundantly describe the transcriptional function of Runx proteins as well as their oncogenic roles as in tumor cells. As the reviewer noticed, we specifically aimed to focus on Runx proteins in the context of immune cell function. We hope the reviewers to find that our review will provide a rather unique view of Runx proteins as modulators of immune cell function.
- “the RUNX family members can only exert their function in transcriptional regulation
by heterodimerization with their non-DNA binding partner protein, core binding factor
beta.” -Is this really true? The Huang paper only talks about protection from
degradation…
We agree with this reviewer that we only have a reference to support the point in which CBFb helps the stabilization of Runx proteins. In the revised manuscript, we added a reference to support the point in which CBFb also helps Runx proteins to bind DNA. And removed the word “only”.
- When talking about the roles of RUNXs in tumorigenesis, it may be a good idea to
mention translocations.
We truly thank the reviewer for this suggestion. However, we respectively disagree with this to describe the roles of Runx proteins in tumorigenesis in this review.
- Page2 “Generally, T cells express RUNX1 and RUNX3, as each of them distinctly
regulates every aspect of their development and effector function. “ - saying every
aspect is quite ambitious..
We agree with the reviewer and have now changed “every” to “many”.
- Page3 Figure 2 has minor issue in the top left corner.
We thank the reviewer for noticing this issue. We have added figure numbers at the top left corner to help the publishing team. We realize that the figures we submit will be published as they are, so we corrected this raised issue.
- Page4 “RUNX1 is required to induce the transactivation of Rorgt locus to execute
the Th17-lineage developmental program but this is counteracted by T-bet, through interaction
with RUNX1 and blockade of RUNX1-mediated transactivation of Rorgt
expression (Lazarevic et al. 2011).” – locus name is Rorc, RORγt – is protein
We thank the reviewer for noticing this mistake. We have modified the text accordingly.
- Page4 “Additionally, Th17 cells are frequently identified in human tumors (Zou and
Restifo 2010; Vitiello and Miller 2020; Bailey et al. 2014).but, unlike the clear roles of
Th1 cells in regulating anti-tumor immunity, the direct roles of Th17 cells in anti-tumor
responses remain elusive. “
We thank the reviewer for highlighting this sentence with a punctuation error (“.but,”). We have now modified the text.
- Page4 “One of the potential reasons why it has been difficult to study Th17 function
in anti-tumor immune responses is the fact that the individuality of Th17 cells is rather
plastic”. – the term “cells individuality” is not clear
We apologize to the reviewer for was not writing this clearly. Our aim was to say that the differentiation of Th17s not fixed by one lineage pathway but rather heavily influenced by various external factors that allow them to adopt various phenotypes such as Th1-Th17s or Treg-Th17. Therefore, Th17s are classified into various subgroups (or subtypes) they can easily transit between them. We have changed the text “the individuality of Th17 cells is rather plastic” to “the identity of Th17 cells is not developmentally fixed.”
- Page4 “Interestingly, Th1-like Th17 cells co-express RUNX1 and RUNX3 and they
both bind to the ROREs in a T-bet-dependent manner to regulate Il17a transcription,
which is in converse to that in Th1 cells, which specifically express RUNX3 to control Tbet-
mediated Ifnγ expression (Wang et al. 2014) “– the sentence is very bulky and a bit
confusing
We agree with this concern raised by the reviewer; we have now rewrote and split this bulky sentence in 2 sentences. Our corrected sentences are the following: “Interestingly, Th1-like Th17 cells co-express RUNX1 and RUNX3 and both together bind to the ROREs to regulate Il17a transcription. Contrastingly, conventional Th1 cells specifically express RUNX3 to control Ifnγ expression in a T-bet dependent manner (34).”
- The last paragraph of part2 on the page4 is predominantly about Th17 biology and
concludes the role of Th17 subset in tumorogenesis rather than about RUNX role. In the
similar manner Page 5 paragraph one mostly discusses CTL cells origin and just a few
words about RUNX. Same for Treg and ILC chapters.
We agree with the reviewer that the descriptive roles of Runxs in these sections is insufficient in Th17 and CD4 CTLs. Unfortunately, there is little literature illustrating the roles of Runxs in tumor infiltrating Th17 and CD4 CTLs and we believe that researchers in the Runx field should focus on these issues, which is one of our motivations for writing this review. Regardless, Th17 and CD4 CTLs have important roles in regulating anti-tumor immunity and Runxs have physiological roles in their development and function, which allow us to speculate their function in response to tumor cells and raise important questions as to how Runx could modulate their anti-tumor immune responses.
- Page5 “Some studies have specifically pointed out the possibility of CD4 CTL to
branch out from a pool of CD4 Th1 cells and have shown the importance of T-bet and
RUNX3 in controlling CD4 CTL differentiation (Takeuchi et al. 2016) as well as the
involvement of ThPOK-Runx3 axis in further mediating this process (Serroukh et al.
2018)”. – this huge sentence can be split by two.
We accept the reviewer’s comment and modified as the following; “Some studies have suggested the possibility of CD4+CTL branching out from a pool of CD4+ Th1 cells and the importance of T-bet and RUNX3 in controlling CD4+ CTL differentiation (44). The ThPOK-Runx3 axis was also shown to be involved in mediating CD4+ CTL differentiation process by the cross-regulating each other (45).”.
- Page5 “Collectively, the RUNX3 transcription factor can strongly influence anti-tumor
immunity of CD4 CTLs via direct or indirect pathways. “
We thank the review and have grammatically corrected this sentence.
- Page5 “Resolution of CTL immune responses results in the generation of immunological
memories, in the forms of memory T cells, which provide a long-term protection against the
same antigenic stimulation and come in 3 subsets. “
We thank the review and have corrected this sentence, as the following: “Resolution of CTL immune responses results in the generation of memory T cells, which provide a long-term protection against the same antigenic stimulation and come in 3 subsets.”
- Page6 “Furthermore, it was mechanistically revealed that RUNX3 controls the
expression of genes responsible for the tissue-residency of CD8 TRM cells as Runx3
downregulation severely promoted tumor growth”. – reference?
We have now included the appropriate reference in the revised manuscript.
- Page7 RUNX1 has been shown to further participate in this pathway by interacting
with FOXP3 to halt the Th17 developmental program, but the molecular mechanism for
RUNX1 cooperation with FOXP3 remains unclear (Zhang, Meng, and Strober 2008) -
Ono et al 2007 showed by co-immunoprecipitation that RUNX1 physically interacts with
FOXP3.
We are glad that the reviewer raises an interesting point and acknowledge the findings of this publication. In the revised version, we included the Ono et al 2007 paper in reference. Our aim was to say that the molecular function for Runx/FoxP3 protein complexes in regulating the Th17 transcriptional program remains unclear as no one has examined their gene targets in Th17s. Therefore, we propose to settle this issue by citing this literature and changing the text from “the molecular mechanism” to “molecular nature”.
- The review lacking recent information about RUNX as a prognostic marker in tumor
infiltration https://pubmed.ncbi.nlm.nih.gov/34485309/
https://pubmed.ncbi.nlm.nih.gov/36092942/
We thank the reviewer for informing these new literature. Again, for the purpose of this review, we wish to focus on Runx’s function only in immune cells and are reluctant to discuss the roles of Runx proteins in tumor cells. We hope that the reviewer understands the scope of our review.
The following phrases are awkward and need to be re-worded:
Deficiency in any of the RUNX family member have detrimental effects on mortality,
These pro-founding studies illustrate
We have now reworded these phrases to the following:
“Deficiency in any of the RUNX family member in mice have detrimental effects on viability…”
“These significant studies illustrate….”

Reviewer 2 Report
Seo et al. provide a global review of the roles of RUNX proteins in mature lymphocytes, especially T cells, natural killer (NK) cells, and Innate Lymphoid Cells (ILCs), with an emphasis on the roles of these cells that may contribute to opposing cancer. Elsewhere, RUNX factors have mostly been reviewed for their roles in cancer as genes that participate in the malignant transformation itself, when they are mis-expressed, mutated, or translocated. This paper, however, covers their many roles supporting the lymphocyte programs that enable cancer to be blocked. Therefore, this paper is a valuable resource that brings together material that is largely new to this reviewer.
The material covered in the review is of great interest. However, the presentation is not ideal in its current form and could be enhanced in several ways. The following points should be considered by the authors to make their work clearer and higher impact for the readers.
1. The authors assume too much knowledge in their readers at the beginning. They throw the reader directly into the complex world of different CD4 cell subsets without ever introducing the difference between CD4 and CD8 T cells in general, or the crucial roles of Runx3 in CD8 T cell development, which was a major discovery by the senior author himself. They also assume that readers will capture the significance of transcription factors like T-bet, Th-POK and ROR gamma t the first times that they are introduced. However, since many cancer biologists will want to read this review and many of them will not be immunology experts, it would improve the paper very much to insert a short introductory section between sections 1 and 2 to review the different types of cells, their usual functions, their developmental relationships, and the key factors that seem to be required for them. With that addition, a much wider audience will be able to appreciate the details that the authors provide.
2. Please be more explicit about the way that TGF-beta works with RUNX factors. Does TGF-beta, working through SMAD TFs, need to upregulate expression of RUNX factors? Does it cause a post-translational modification of RUNX proteins? Or does TGF-beta-mobilized SMAD bind next to RUNX on the DNA and cooperate with it? Since this relationship is referred to in several different places, it should be clearer what is meant.
3. Throughout the paper, roles of lymphocytes in normal immune responses and in tumor control are discussed as if they are obviously the same thing, often described in a single sentence. However, this is probably not true, since microenvironments inside a tumor are very different from microenvironments in normal tissues or in secondary lymphoid organs. Please make this distinction between roles in normal immune function and specific roles in tumor control clearer. For example, it is thought that the tumor microenvironment is partially responsible for causing CD8 T cell exhaustion, so simply being an active CD8 CTL is not enough to be effective against a tumor. In this example, if there is literature specifically showing that RUNX factors help the T cells avoid exhaustion, for example, that should be singled out for mention as a tumor control-specific role.
4. It is also especially uncertain how the roles of T(RM) cells and other tissue-resident lymphocytes in normal tissue surveillance relate to their roles when the tissue is disrupted by a tumor, either through oncogenic transformation or through metastasis. While a comprehensive discussion isn’t needed, I think the authors need to clarify the distinctions between these roles.
5. Finally, there are some minor writing issues that the authors may want to consider.
a. In a few places, there are mismatches between singular and plural nouns and verbs, e.g. in line 3 of the abstract, on p. 2 on line 7 from the bottom, p. 3 on line 6 from the bottom.
b. On p. 4, please define “TME” here. Right now it is only defined below, on p. 8.
c. On p. 7, do the authors mean that Tregs are essential for maintaining immune “homeostasis” or “quiescence” or “correct activation thresholds”?
d. Could a reference be added for the effect of Tregs in bringing cAMP into effector T cells? (Middle of p. 7)
e. Also, further down on p. 7, T cell exhaustion is indeed cell intrinsic to the CD8 T cells, but saying this here is potentially confusion. What is being described here is the role of Tregs in causing the CD8 T cells to become exhausted. So that effect, one cell to another, is not cell intrinsic.
f. Beginning section 6, better to say “In contrast to CD4 and CD8 T cells…”
g. In section 7 near the end, it is not clear what is meant by the sentence referring to the paper by Seo et al. 2020. Do the authors mean “A novel role of RUNX3 in suppressing…. AND REGULATING anti-tumor immune responses”? Or do they mean “A novel role of RUNX3 in suppressing…. TO REGULATE anti-tumor immune responses”?
Author Response
<To the Reviewer 2>
Elsewhere, RUNX factors have mostly been reviewed for their roles in cancer as genes that participate in the malignant transformation itself, when they are mis-expressed, mutated, or translocated. This paper, however, covers their many roles supporting the lymphocyte programs that enable cancer to be blocked.
We appreciate that the reviewer acknowledges our intention of writing this review.
- The authors assume too much knowledge in their readers at the beginning. They throw the reader directly into the complex world of different CD4 cell subsets without ever introducing the difference between CD4 and CD8 T cells in general, or the crucial roles of Runx3 in CD8 T cell development, which was a major discovery by the senior author himself.
We agree with the reviewer. Please see our response below.
They also assume that readers will capture the significance of transcription factors like T-bet, Th-POK and ROR gamma t the first times that they are introduced. However, since many cancer biologists will want to read this review and many of them will not be immunology experts, it would improve the paper very much to insert a short introductory section between sections 1 and 2 to review the different types of cells, their usual functions, their developmental relationships, and the key factors that seem to be required for them. With that addition, a much wider audience will be able to appreciate the details that the authors provide.
To alleviate this important concern raised by the reviewer, we have added the following paragraph in section 1 (on page 2) to provide more background on CD4 and CD8 T cells .
“At the periphery, CD4 and CD8 T cells activate and differentiate into various effector T cells subsets, with each subset having a unique function to the control clearance of viral, fungal, or bacterial pathogens. Activation of CD4+ naïve T helper (Th) cell requires antigen presentation by MHC-II expressing antigen presenting cells (APC), which allow their dif-ferentiation into multiple effector subsets. The differentiation into each T helper subset is mainly controlled by the cytokines present in their microenvironment. These CD4+ T help-er subsets include: Th1 driven by IL-12: Th2 by IL-4; Th9 by IL-4 and TGF-β; Th17 by IL-6 and TGF-β; Th22 by IL-6 and IL-23; and Treg by TGF-β and IL-2 (13, 14). The cytokines that mediate CD4 T helper differentiation activate the downstream signaling pathways and lead to the upregulation of the master transcription factor (see box 1). Each of these master transcription factors is responsible for defining transcriptional programs for their CD4 T helper subset and finetune the expression of appropriate cytokines to control the immune response. CD8 T cells respond to antigens presented on MHC class I proteins and differentiate into cytotoxic T lymphocytes (CTL), the effector T cells that express cyto-lytic proteins and directly mediate the clearance of virally infected cells and malignant cells.”
In addition, we have supplemented a schematic figure (in a box format) which illustrates the developmental cues for CD4 helper and CTL cells and hope to keep the cancer biologists interested in their function in tumor suppression.
- Please be more explicit about the way that TGF-beta works with RUNX factors. Does TGFbeta, working through SMAD TFs, need to upregulate expression of RUNX factors? Does itcause a post-translational modification of RUNX proteins? Or does TGF-beta-mobilized SMAD bind next to RUNX on the DNA and cooperate with it? Since this relationship is referred to in several different places, it should be clearer what is meant.
We thank the reviewer for the constructive suggestions. We modified the manuscript to add more specific descriptions of the relationship between TGFb and Runxs as follows.
â–ªNotably, RUNX are also activated downstream of TGF-β signaling and cooperate with SMAD signal transducers to regulate the expression of their target genes (12),
â–ªFurther studies in murine gastric epithelium revealed that RUNX3 functions as a tu-mor-suppressor downstream of TGFβ signaling and forms a protein complex with SMADs to induce the expression of the cyclin-dependent kinase inhibitory protein-1 p21waf1/cip1 (19). CIP1p21 inhibits cell cycle progression in responses DNA damage, senescence, growth inhibitory signals, and tumor suppression.
â–ªIn addition, specific environmental signals such as TGFβ and RA (Retinoic acid) are re-sponsible for suppressing ThPOK and promoting RUNX3 in these intestinal CD4+ CTLs..
- Throughout the paper, roles of lymphocytes in normal immune responses and in tumor control are discussed as if they are obviously the same thing, often described in a single sentence. However, this is probably not true, since microenvironments inside a tumor are very different from microenvironments in normal tissues or in secondary lymphoid organs. Please make this distinction between roles in normal immune function and specific roles in tumor control clearer. For example, it is thought that the tumor microenvironment is partially responsible for causing CD8 T cell exhaustion, so simply being an active CD8 CTL is not enough to be effective against a tumor. In this example, if there is literature specifically showing that RUNX factors help the T cells avoid exhaustion, for example, that should be singled out for mention as a tumor control specific role.
We apologize for confusing the reviewer. We agree that the immune responses against pathogens (or allergens) or tumors are very different but we also cannot ignore that there are also similarities between them. Unfortunately, our knowledge of Runxs in immune system responses are mainly derived from studies under non-tumor settings, and there is currently no literature addressing the role of Runx in regulating T cell exhaustion. Therefore, this makes it very difficult for us to differentiate the roles of Runxs in non-tumor versus tumor-related situations (this is in fact the main motivation of writing this review). To resolve the reviewer’s concern, we have added the following sentence at the end of the introduction: “Additionally, we will discuss the roles of RUNX in regulating lymphocyte function under physiological conditions and raise important questions as to how these immune responses could potentially control the host`s antitumor immune responses to cancer. .”
- It is also especially uncertain how the roles of T(RM) cells and other tissue-resident lymphocytes in normal tissue surveillance relate to their roles when the tissue is disrupted by a tumor, either through oncogenic transformation or through metastasis. While a comprehensive discussion isn’t needed, I think the authors need to clarify the distinctions between these roles.
We thank the reviewer for this suggestion. We have now included a sentence at the end of our Trm section (page 7) to reflect this as “In addition, the presence other tissue-resident lymphocytes such as γδT cells, iNKT (in-variant natural killer) cells, MAIT (mucosal-associated invariant T) cells in controlling tumorigenesis could have some influence on the TME. However, whether γδT cells, iNKT cells, MAIT cells require RUNX for their anti-tumor immune responses is unknown.”.
We agree that it is still unclear how the tissue resident lymphocytes respond to the tumors. Unfortunately, there is a lot of heterogeneity in tumors that express a wide array of antigens and ligands and affect the immune cell function, thus obscuring our understandings to how they respond in the TME.
However, one significant study by Thomas Gebhardt and colleges (Nature, 2018) demonstrated the importance of TRMs in tumor immune surveillance of melanoma and promote cancer–immune equilibrium to halt cancer progression: mice deficient in TRMs formation were more susceptible to tumor development; depletion of TRMs triggered tumor growth; and most striking of all, injection of TRMs alone in melanoma bearing Rag1 KO mice (mice that are deficient in B and T cells) was sufficient to suppress tumor development. Therefore, this really put TRMs at the forefront of the host`s immune response to cancer.
To clarify TRMs roles in tumors, we have now included a sentence on page 6 so say that TRMs are critical regulators of cancer–immune equilibrium and protect against epidermis-derived cancers.
- Finally, there are some minor writing issues that the authors may want to consider.
- In a few places, there are mismatches between singular and plural nouns and verbs,
e.g. in line 3 of the abstract, on p. 2 on line 7 from the bottom, p. 3 on line 6 from the bottom.
We thank the reviewer for identifying mistakes in our manuscript. We reexamined the text and fixed the issues.
- On p. 4, please define “TME” here. Right now it is only defined below, on p. 8.
We thank the reviewer and have now provided the definition of TME on page 4.
- On p. 7, do the authors mean that Tregs are essential for maintaining immune
“homeostasis” or “quiescence” or “correct activation thresholds”?
We apologize for not clearly defining the roles of Tregs. We have changed “Tregs are essential for maintaining immune homeostasis” to “Tregs are essential for maintaining self-tolerance and regulating immune homeostasis.”
- Could a reference be added for the effect of Tregs in bringing cAMP into effector T
cells? (Middle of p. 7)
We have now added the reference.
- Also, further down on p. 7, T cell exhaustion is indeed cell intrinsic to the CD8 T cells, but saying this here is potentially confusion. What is being described here is the role of Tregs in causing the CD8 T cells to become exhausted. So that effect, one cell to another, is not cell intrinsic.
We thank the reviewer for this important suggestion. We removed the word “intrinsic”.
- Beginning section 6, better to say “In contrast to CD4 and CD8 T cells…”
We have now modified the text accordingly to this suggestion.
- In section 7 near the end, it is not clear what is meant by the sentence referring to the
paper by Seo et al. 2020. Do the authors mean “A novel role of RUNX3 in suppressing…. AND
REGULATING anti-tumor immune responses”? Or do they mean “A novel role of RUNX3 in
suppressing…. TO REGULATE anti-tumor immune responses”?
We apologize for confusing the reviewer. We meant the latter. To further clarify this, we modified the text as “We uncovered a novel anti-tumor role of RUNX3 in lung-resident NK cells through sup-pressing Ccl5 expression”.
